# Variable Speed Limit Control for the Motorway–Urban Merging Bottlenecks Using Multi-Agent Reinforcement Learning

Xuan Fang *, Tamás Péter and Tamás Tettamanti

Department of Control for Transportation and Vehicle Systems, Faculty of Transportation Engineering and Vehicle Engineering, Budapest University of Technology and Economics, Műegyetem rkp. 3, H-1111 Budapest, Hungary; peter.tamas@kjk.bme.hu (T.P.); tettamanti.tamas@kjk.bme.hu (T.T.)
* Correspondence: fangxuan@edu.bme.hu

**Abstract:** Traffic congestion is a typical phenomenon when motorways meet urban road networks. At this special location, the weaving area is a recurrent traffic bottleneck. Numerous research activities have been conducted to improve traffic efficiency and sustainability at bottleneck areas. Variable speed limit control (VSL) is one of the effective control strategies. The primary objective of this paper is twofold. On the one hand, turbulent traffic flow is to be smoothed on the special weaving area of motorways and urban roads using VSL control. On the other hand, another control method is provided to tackle the carbon dioxide emission problem over the network. For both control methods, a multi-agent reinforcement learning algorithm is used (MAPPO: multi-agent proximal policy optimization). The VSL control framework utilizes the real-time traffic state and the speed limit value in the last control step as the input of the optimization algorithm. Two reward functions are constructed to guide the algorithm to output the value of the dynamic speed limit enforced within the VSL control area. The effectiveness of the proposed control framework is verified via microscopic traffic simulation using simulation of urban mobility (SUMO). The results show that the proposed control method could shape a more homogeneous traffic flow, and reduces the total waiting time over the network by 15.8%. In the case of the carbon dioxide minimization strategy, the carbon dioxide emission can be reduced by 10.79% in the recurrent bottleneck area caused by the transition from motorways to urban roads.

**Keywords:** variable speed limit; reinforcement learning; multi-agent proximal policy optimization; road traffic control; traffic emission

## 1. Introduction

As an important transportation infrastructure, motorways represent the overall level of a country's transportation system to a large extent and play a vital role in the development of the national economy. However, motorways are also facing increasingly frequent traffic congestion [1]. As a road section that is prone to generating and spreading congestion, the traffic bottleneck area is an important research object to improve motorway traffic management capabilities. According to the specific time and location of occurrence, traffic bottleneck areas can be classified into recurring and nonrecurring bottleneck areas [2]. Recurring traffic bottleneck areas are the confluence area, diversion area, and weaving area formed when the main line of the motorway merges with the entrance ramp and exit ramp. Congestion frequently occurs in recurring traffic bottleneck areas [3]. The closure of lanes due to road construction, bad weather, or traffic accidents has stochastic characteristics, which are the causes of nonrecurring bottleneck areas [4–6].

Compared with motorways, the distribution of entrance and exit ramps at the merging area of motorways and urban roads is more concentrated, and adjacent entrances and exits are more closely connected with urban roads, resulting in more traffic conflicts in the merging area of motorways and urban roads [7,8]. Compared with urban roads, the

merging area of motorways and urban roads often service more traffic demand due to the closed traffic environment and connectivity of settlements and cities, resulting in a broader range of traffic bottlenecks and faster spread speed [9,10].

As a part of the intelligent transportation system (ITS), variable speed limit (VSL) control has been widely used as one of the traffic control measures to improve traffic efficiency [11–13], benefit the environment [14,15], and enhance traffic safety [16,17] in bottleneck areas. By adjusting the speed limit of the main line upstream of the bottlenecks, the number of vehicles entering the motorway bottleneck area during the congestion period is controlled within a certain range to keep the traffic state more uniform and stable. VSL control methods in traffic management can be divided into different categories according to their basic approaches and techniques. The most used VSL control strategies are rule-based approaches [18,19] and model-based approaches. Rule-based VSL control approaches use a set of predefined rules and thresholds to determine the appropriate speed limits. These rules take into account factors such as traffic density, volume, occupancy, and historical data to set speed limits. Rule-based VSL approaches may cause traffic flow fluctuations when the traffic state exceeds or falls below the set thresholds. Model-based VSL control approaches utilize mathematical models to represent and predict traffic behavior. Model-based approaches using optimization algorithms to determine the optimal speed limits. Model-based approaches are further classified into open-loop optimization approaches [20–25] and feedback control approaches [26–28]. Model-based approaches require accurate models to describe the state of traffic flow. Due to the need for a large number of numerical calculations and the fact that the model contains many parameters to be calibrated, the model-based VSL control approaches have limitations in large-scale traffic control applications. In addition, the portability of these VSL control approaches needs further work. This means when applying the control algorithm to a new scenario, the traffic flow model needs to be re-calibrated, and the controller parameters need to be adjusted manually.

With the development of artificial intelligence technology, VSL control based on reinforcement learning (RL) can automatically adapt to various traffic environments and achieve optimal control effects without specific traffic flow models [29–31]. Existing RL-based VSL control approaches use the matured deep Q-network (DQN) developed in 2015 [32]. The continuous development of reinforcement learning research has brought new solutions to RL-based VSL control problems.

This study proposes a fully cooperative multi-agent reinforcement learning framework to solve the multi-section VSL control optimization problem at the merging area of motorways and urban roads. This framework is based on the multi-agent proximal policy optimization (MAPPO) algorithm proposed in 2021 [33], which has been proven powerful for target localization [34], production scheduling [35], and trajectory planning [36] but has not been utilized to solve the VSL optimization problem. Under the constraints of rational speed, traffic flow state information, action representing discrete speed limit values, and reward composed of occupancy data and carbon dioxide emissions are designed to smooth traffic and reduce emissions. The proposed multi-agent reinforcement learning framework is tested on the microscopic traffic simulator simulation of urban mobility (SUMO) [37].

## 2. Methodology

This study presents a VSL control framework, applying multi-agent reinforcement learning (MARL) with the proximal policy optimization (PPO) algorithm. Section 2.1 introduces the literature background of RL. Section 2.2 presents the basis of the PPO algorithm. Section 2.3 extends the basic PPO algorithm to MARL.

### 2.1. Reinforcement Learning

Reinforcement learning (RL) is a machine learning (ML) algorithm. During the learning process, the agent obtains the optimal strategy by trying different action choices and adjusting the evaluation value of the action according to the feedback of the environment.

During each interaction between the agent and the environment, the input to the agent is the environment state *s*. The agent chooses action *a* as the output to transit the state of the environment to state *s'*, and at the same time, the agent receives reward *r*. RL aims to find a series of optimal action sets [38]. Deep learning (DL) is a method based on representational learning of data in ML, which abstracts the features contained in the data based on the value of the original data for representation [39]. The core of DL is the deep neural network (DNN), inspired by the principles of neural networks (NNs) in biology. NN abstracts the structure of the human brain and the response mechanism to external stimuli in a mathematical model.

Deep reinforcement learning (DRL) is an algorithm that combines RL with the ability to learn from the environment and DL with powerful representation capabilities. DRL has two branches: Q-learning and policy gradient (PG) algorithms [40]. Q-learning is mainly based on the iteration of the action–value function to model the value of the state space to find the optimal strategy. The Q-learning algorithm has excellent interpretability and debuggability but relies on extensive experience sample storage to establish an accurate model. PG directly outputs the probability of the action. Since no additional state–action value is introduced, PG requires fewer hyperparameters to be tuned, and due to the gradient method utilized, PG will update toward the direction of the optimization strategy, which means it has good convergence. But the disadvantage is that PG can easily converge to a local optimal value.

The actor–critic (AC) algorithm [41] combines the advantages of Q-learning and PG. AC uses the single-step update advantage of Q-learning to allow the Q-network as a critic to learn offline (i.e., an agent that learns through experience samples that are not acquired by itself). The critic provides an evaluation of the actor's action or the direction of the gradient descent in a single-step policy update, which greatly improves the update frequency of the actor and accelerates the learning rate. In this way, the problem of the low efficiency of the PG in episode updates can be solved, but there is a problem of convergence difficulty. To solve the difficulty of convergence of the AC algorithm and speed up training, the asynchronous advantage actor–critic (A3C) algorithm [42] pushes the AC algorithm into multi-threads for synchronous training. The emerging trust region policy optimization (TRPO) algorithm [43] solves the problem of fluctuations in the A3C algorithm when balancing the variance and bias of the model.

*2.2. Proximal Policy Optimization*

Based on PG and TRPO, the proximal policy optimization (PPO) algorithm [44] is proposed as an improved method. The PPO algorithm solves the shortcomings of previous RL algorithms, such as low data utilization efficiency, poor robustness of the PG algorithm, and the complexity of the TRPO algorithm. The PG algorithm is based on calculating the estimator of the policy gradient and substituting it into the stochastic gradient ascent algorithm. The unbiased estimate of the gradient is

$$\delta \widehat{J}(\theta) = \widehat{E}_t[\delta_\theta log \pi_\theta(a_t|s_t)\widehat{A}_t] \tag{1}$$

where $\pi_\theta$ is a random strategy for actor network parameters $\theta$; $\widehat{A}_t = Q_t - V_t$ is the advantage function estimated by critic; $Q_t$ is the state–action value function; and $V_t$ is the state–value function. $a_t$ is the action space, and $s_t$ is the state space. However, PG needs to re-interact with the environment after each update of the parameters, calculate the advantage function of the new strategy, and then update the parameters, making the update speed slow. The TRPO algorithm uses the importance sampling method. TRPO adopts the new and old strategy networks $\theta$ and $\theta_{old}$ to allow the network to interact with the environment and use the collected data to train the network $\theta$ so that the agent can perform multiple parameter updates in one interaction with the environment, which improves the update speed.

In addition, to prevent the gap between the new network $\theta$ and the old network $\theta_{old}$ from being too large, the relative entropy is used to measure the difference between the two, then the objective function is

$$\max_{\theta} J(\theta) = \widehat{E}_t \left[ \frac{\pi_\theta(a_t|s_t)}{\pi_{\theta_{old}}(a_t|s_t)} \widehat{A}_t \right],$$

$$\text{subject to } \widehat{E}_t[KL[\pi_{\theta_{old}}(a_t|s_t), \pi_\theta(a_t|s_t)]] \leqslant \zeta \tag{2}$$

where $\zeta$ is the radius of the trust domain, and *KL* is the boundary conditions. The PPO adds the *KL* of TRPO as a penalty item to the objective function and applies an adaptive penalty of the $\beta$ parameter to the *KL* divergence. The actor's objective function is

$$\max_{\theta} J^{KL}(\theta) = \widehat{E}_t \left[ \frac{\pi_\theta(a_t|s_t)}{\pi_{\theta_{old}}(a_t|s_t)} \widehat{A}_t - \beta D_{KL}[\pi_{\theta_{old}}(a_t|s_t)||\pi_\theta(a_t|s_t)] \right] \tag{3}$$

where $D_{KL}(\pi_{\theta_{old}}||\pi_\theta)$ represents the divergence of *KL* between the $\pi_{\theta_{old}}$ and the $\pi_\theta$ controls the difference in each episode of policy updates. $\beta$ is adaptively adjusted according to the preset *KL* divergence threshold. In practice, the clip function is used to limit the probability ratio $r_t(\theta)$ to $(1 - \varepsilon, 1 + \varepsilon)$, where $\varepsilon$ is a hyperparameter, also taking $\frac{\pi_\theta}{\pi_{\theta_{old}}} \widehat{A}_t$ as the optimization object. When $\widehat{A}_t > 0$, increase the probability of $\pi_\theta(a_t|s_t)$ and vice versa. The actor's objective function is

$$\max_{\theta} J^{clip}(\theta) = \widehat{E}_t \left[ clip\left( \frac{\pi_\theta(a_t|s_t)}{\pi_{\theta_{old}}(a_t|s_t)}, 1 - \varepsilon, 1 + \varepsilon \right) \widehat{A}_t, \frac{\pi_\theta(a_t|s_t)}{\pi_{\theta_{old}}(a_t|s_t)} \widehat{A}_t \right] \tag{4}$$

The optimization framework of the PPO algorithm for traffic control is shown in Figure 1.

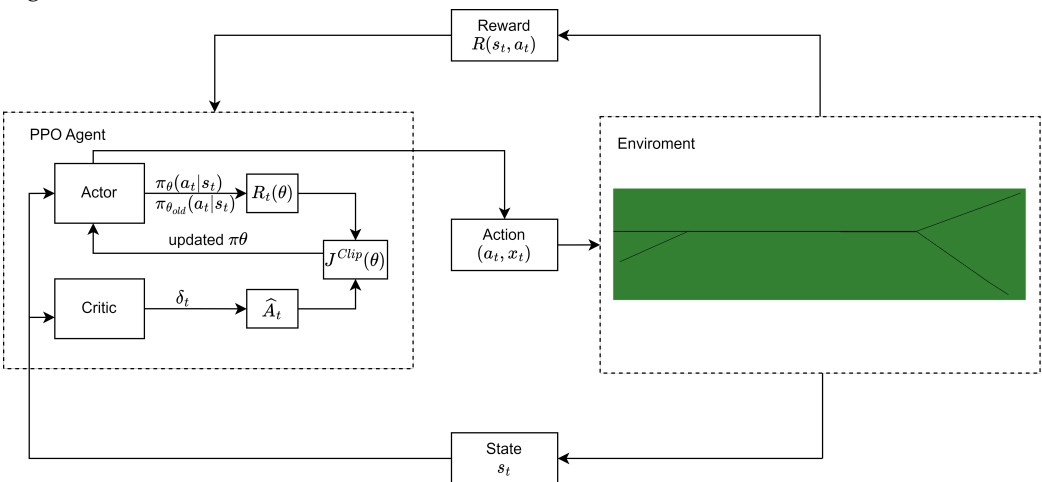

**Figure 1.** PPO agent optimization framework.

### 2.3. Multi-Agent Proximal Policy Optimization

An individual agent is inadequate to handle multi-section VSL control. For this reason, the multi-agent reinforcement learning (MARL) framework is developed. Based on the centralized training with decentralized execution (CTDE) framework, the PPO algorithm is extended to the MAPPO algorithm. That is, each individual PPO agent is trained using a global value function. After the training is completed, each individual PPO agent obtains a decentralized strategy, and action is taken locally based on this strategy. Then, centralized training is applied to make the decentralized strategies learned by individual PPO agents work cooperatively. In the fully cooperative MAPPO algorithm, all individual PPO agents share common reward signals. During centralized training, a global critic network is used to evaluate the state value and calculate $\widehat{A}_t^i$ using the generalized advantage estimation (GAE)

method. During decentralized execution, each individual PPO agent relies on its local observation state to realize a distributed decision-making interactive environment. The global critic network uses GAE to estimate a common $\widehat{A}_t$ based on the reward $R_t^i$ obtained by each agent's action trajectory during centralized training. The objective function for agent $i$ is

$$\max_{\theta} J_i(\theta) = \widehat{E}_t \left[ clip \left( \frac{\pi_\theta(a_t^i|s_t^i)}{\pi_{\theta_{old}}(a_t^i|s_t^i)}, 1 - \varepsilon, 1 + \varepsilon \right) \widehat{A}_t^i, \frac{\pi_\theta(a_t^i|s_t^i)}{\pi_{\theta_{old}}(a_t^i|s_t^i)} \widehat{A}_t^i \right] \tag{5}$$

## 3. Simulation Environment

Unlike other control algorithms, the DRL algorithm does not directly learn the model from a given data set but learns by interacting with the environment continuously to generate data. Implementing the training of the algorithm directly in the real traffic network is not feasible, which will bring huge costs and safety hazards. The usual feasible method is to train the RL algorithm with the traffic simulation platform and then migrate the trained model to the real traffic network. In addition, the multi-agent RL framework requires an environment to interact. To implement the proposed multi-agent reinforcement learning framework, an accurate and efficient traffic simulation platform is essential. The simulation of urban mobility (SUMO) is an open-source, multi-modal traffic simulator that can be extended dynamically and is highly customizable through the embedded traffic control interface (TraCI).

To test the control performance of the proposed multi-agent reinforcement learning framework, a dynamic traffic network representing the merging area when the motorway transited to the urban roads i sgenerated based on the OpenStreetMap (OSM) data. Figure 2 shows the schematic diagram of the proposed MAPPO-based VSL control at the typical recurrent bottleneck area located on the border of Budapest in Hungary. The intersection with the ramps and the existence of traffic lights in the urban road network leads to frequent lane-changing behavior of vehicles in the merging area, which further leads to congestion.

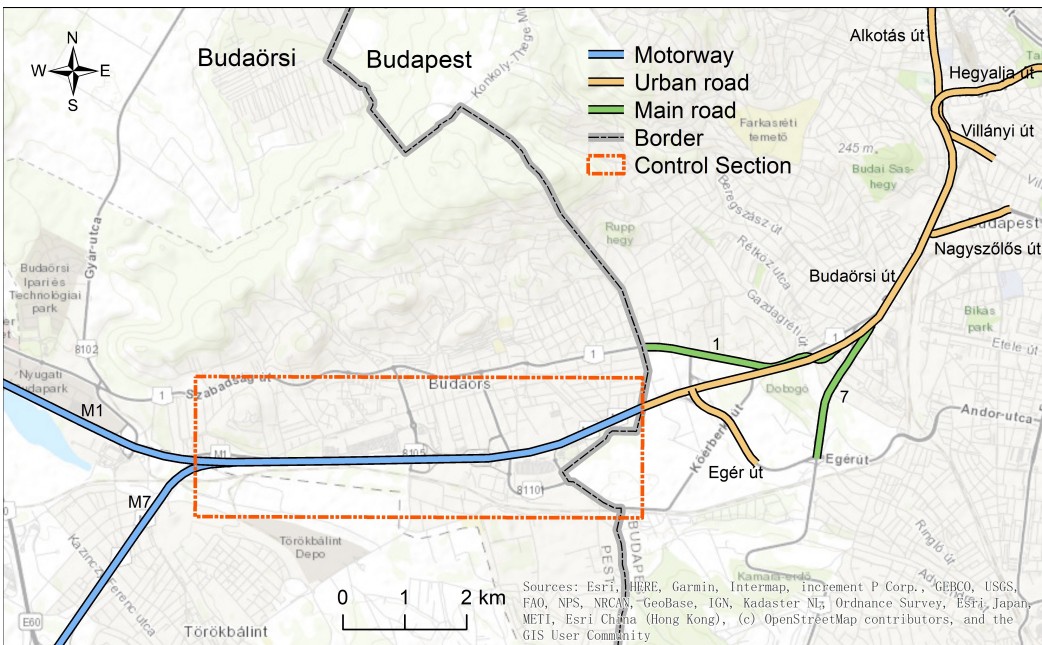

**Figure 2.** Schematic diagram of variable speed limit control at the merging area, where the motorway transitions to the urban road network.

The geometric structure of this bottleneck area in SUMO is shown in Figure 3. The control section is divided into seven homogeneous sections, named "section_a", "section_b", "section_c", "section_d", "section_e", "section_f", and "section_g", each of which is one

kilometer long. There are four vehicle routes in the road network. Two of them start from the motorways and end on the urban roads Alkotás Street and Hegyalja Street, while the other two are the on-ramp flow starting from Egér Street. According to on-site historical measurement data, traffic flows are assigned to the road network at a rate of 2/3, with more vehicles traveling to Alkotás Street. The total demand on the main road is 4657 veh/h and 1039 veh/h on the ramp. Two traffic light control systems exist at the end of this regional network. The control schema is the fixed cycle with 90 s cycle time. The green phase is 60 s, and the yellow phase is 3 s.

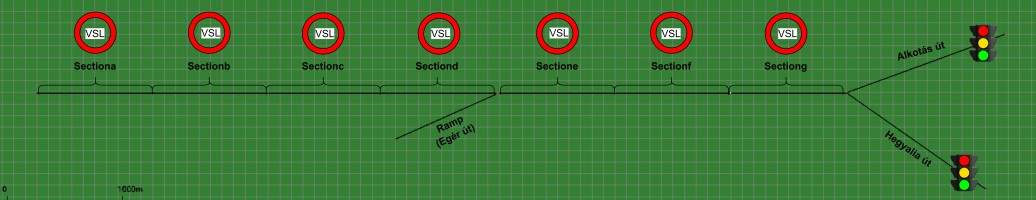

**Figure 3.** Geometric structure of simulated recurrent bottleneck area.

To connect the dynamic traffic environment in SUMO with the MAPPO algorithm, a custom environment "SUMO_Env()" is created in PyCharm, which contains the main subset functions needed for the training, initialization simulation, state representation, action space, reward, termination, reset, and step functions.

The initialization of the simulation function initializes the settings of the simulation environment. The simulation time is set to 7200 s. The original speed limit of the road network is set to 27.78 m/s (100 km/h) for motorways and 13.89 m/s (50 km/h) for urban roads according to the actual traffic rules.

The states obtained by RL agents in different environments are various. For example, in the robot simulation environment, the state representation is mainly composed of joint dynamics data in the physical sense [45]. In the game task, the state information agents obtained from the environment are almost exclusively image data [46]. In financial decision tasks, the state information includes stock market bonds, various K-line indicators, transactions, and financial data [47]. Detailed state representation helps RL algorithms extract critical information. The state-space representation in this paper is the vehicle occupancy data in each section, including main roads and ramps.

The action space represents the actions of the PPO agents applied to the traffic environment. Six discrete speed limit values make up the action space in this study for the dynamic traffic environments, which is $[36.11, 30.55, 25, 19.44, 13.89, 8.3]$, the unit is m/s. The goal of agents is to cooperate with each other to find the optimal speed limits set for different road sections.

The reward function setting is very critical in RL problems. The reward function represents the optimization goal of the training. It is the only feedback from the environment to the actions that the agent takes. Commonly used reward functions in RL-based VSL control are total time spent (TTS) [30,48], density distribution [12,49], mean speed in the bottleneck [50] that is utilized to improve traffic efficiency and the negative sum of the values of tail gas emissions [50–52] that take the environmental sustainability into account. In this paper, two reward functions are introduced to smooth the traffic flow and reduce traffic emissions. The first reward function is the negative of the standard deviation of the vehicle occupancy data on different road sections. The second function is the negative sum of the scaled values of carbon dioxide emissions.

The termination of training depends on the training iteration and simulation. When the training iteration reaches the preset value and the simulation time reaches the predefined value, the connection between the MAPPO framework and SUMO simulation is closed.

The reset function is responsible for resetting the environment to its initial state before the start of a new episode. It sets the state of each agent in the environment to its initial state.

The step function executes a single step in the environment for each PPO agent and collects the resulting observations, rewards, and speed limits. The step function receives the actions chosen by each individual PPO agent and applies them to the environment. After applying the actions, the step function collects the current state of the environment, which is used as input for the agent's policy update. The step function calculates the rewards obtained by each agent based on the predefined reward function. The step function determines if the current state of the environment is a terminal state, indicating the end of the episode. The step function also collects the speed limits chosen by the agents, which can be considered the control input and can be used for further analysis.

## 4. Results and Discussion

In the sequel, the simulation results are introduced in detail, and relevant discussion is provided as well.

### 4.1. Training Setting of MAPPO Algorithm

To select the appropriate variables for the training process, hyperparameters need to be properly tuned. The set of MAPPO hyperparameters utilized in this paper is shown in Table 1.

**Table 1.** Multi-agent proximal policy optimization hyperparameters for training.

| Hyperparameters | Value |
| --- | --- |
| Number of training iterations | 600 |
| Learning rate | 0.0005 |
| Number of agent | 7 |
| PPO clip parameter $\theta$ | 0.2 |
| Discount factor $\gamma$ | 0.99 |
| GAE $\lambda$ parameter | 0.95 |
| Time step per update | 120 |
| Number of PPO epochs per update | 15 |
| Hidden layers | $64 \times 64 \times 64$ |
| Hidden layers activation function | RELU |

The training performance is verified by the reward curve through 600 iterations. In each iteration, a random seed is applied for the proposed control strategies to keep the diversity. Seven PPO agents are used in this paper, and each agent controls a road section; they work cooperatively to minimize the reward function. The GAE $\lambda$ parameter represents the parameter used in GAE methods that controls the trade-off between bias and variance in the advantage estimation. The number of PPO epochs per update determines how many times the network parameters are updated based on the collected data during a single training iteration. The number of hidden layers in the neural network is 3, the number of neurons in the hidden layer is 64, and the activation function of the hidden layer is the ReLu function, which helps with the vanishing gradient problem, allows the network to learn non-linear relationships, and provides sparse activation.

Traffic performance and sustainability measurements are taken in a dynamic traffic environment based on the control output obtained by different reward settings. The traffic performance measurement, including the occupancy data of each road section and its distribution, shows the effect of the proposed control framework in smoothing the traffic. The sustainability measurements are presented by the reduction in carbon dioxide emissions.

### 4.2. Traffic Performance Measurements

Figure 4 shows the learning curve over 600 iterations, utilizing the negative standard deviation of the vehicle occupancy of different road sections as the reward function. Min-Max normalization is performed on the obtained rewards. The reward value changes significantly when the number of training iteration reaches about 76. Although there are some oscillations, the reward value is generally stable in the subsequent training process.

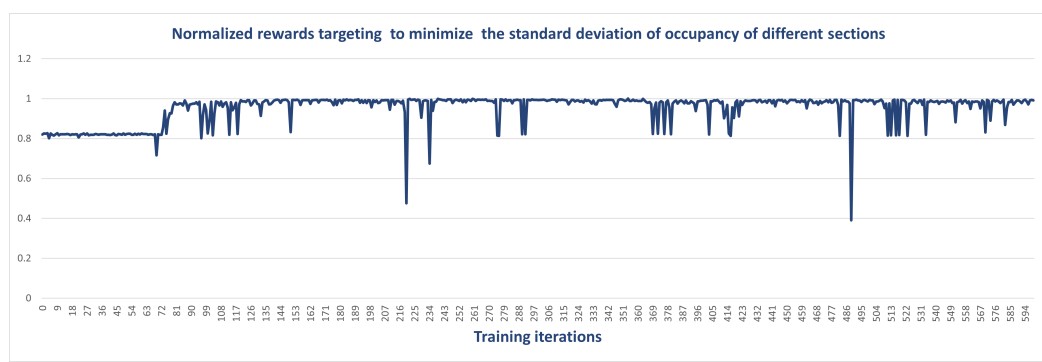

**Figure 4.** The learning curve dedicated to smooth traffic flow over the 600 iterations.

As Figure 5 shows, the MAPPO-based VSL controller starts to reduce the speed limit of all controlled sections at step 25. The speed limit is stable between 8.3 m/s and 13.89 m/s after 104 steps. When traffic congestion is about to be formed at the bottleneck area, the MAPPO-based VSL controller reduces the speed limit of the controlled road sections quickly to 13.89 m/s to prevent capacity drop at the bottleneck area. In this paper, we assume that drivers' acceptance of the speed limit is 100%. The proposed speed limit can be spread to the road network in a traditional way, i.e., variable speed limit signs. With the connected and automated vehicles (CAVs) and intelligent transportation infrastructure on the road network, the proposed speed limit can be synchronously received by CAV with vehicle-to-vehicle (V2V) communication technology.

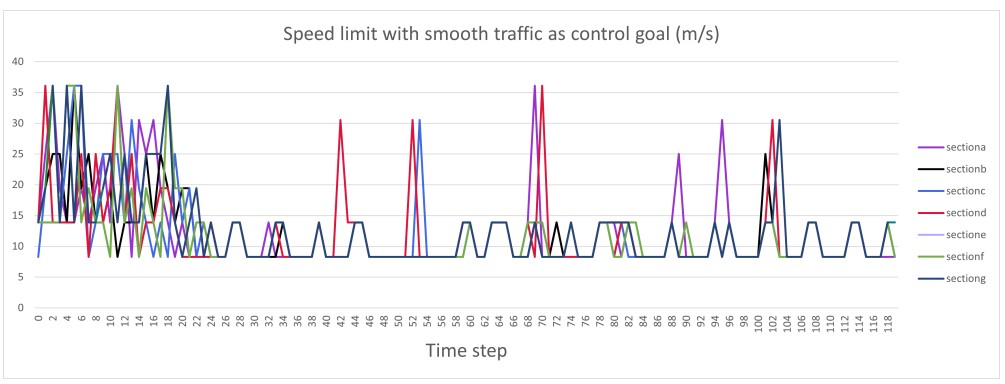

**Figure 5.** The speed limit of different road sections with smooth traffic as the control goal.

The changes in the occupancy data of different road sections during the simulation period without VSL control are shown in Figure 6. The first 300 s is the warming-up time. The occupancy change shows that from 2400 s to the end of the simulation, the occupancy of the motorway sections before merging with the ramp varies between 36% and 68%. With the proposed MAPPO-based VSL control, as shown in Figure 7, the occupancy of the motorway sections before merging with the ramp and "section_f" reaches almost the same value of 46% at the end of the simulation. The occupancy of "section_e" and "section_g" reaches the same value of 30%. From this, a conclusion can be made that the controller truly achieves the predefined goal, which is that by controlling the speed limit of each road section, the vehicles are evenly distributed on each road section.

Figure 8 intuitively demonstrates the distribution of vehicles on the road network without VSL control. It shows that vehicles queued at "section_a", "section_b", "section_c", and "section_e". This was caused by merging with the ramp and the backpropagation of congestion occurring in the urban area. After implementing MARL-based VSL control, as shown in Figure 9, vehicles began distributing uniformly and concentratedly on all motorway sections. This is because the PPO agents started to reduce the upstream speed limit to obtain the optimal reward defined by the agent policy. The interquartile range of the box plot shows how the data fluctuate. Compared with the non-implementation of

the VSL control case, occupancy fluctuations in the upstream section of the motorway are significantly smaller when applying the MARL-based VSL control.

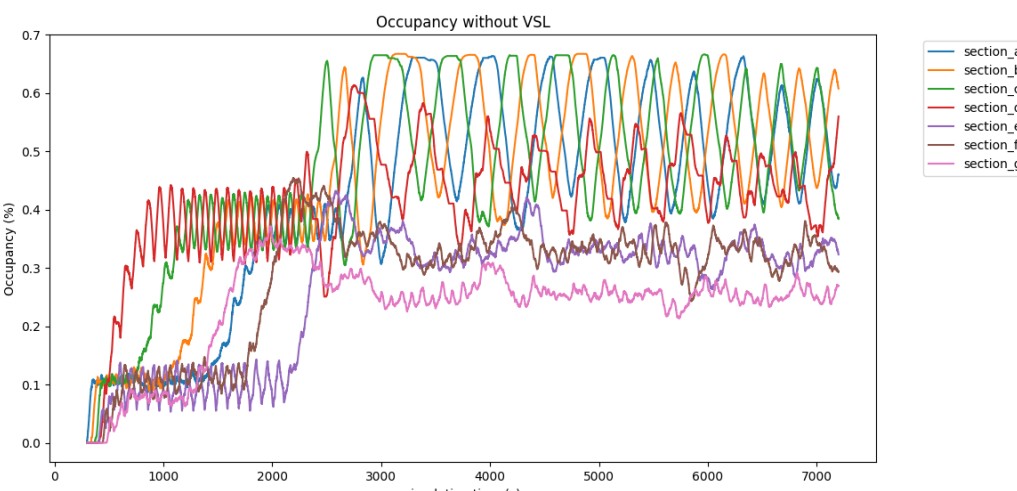

**Figure 6.** Occupancy data of different road sections during the simulation period without VSL control.

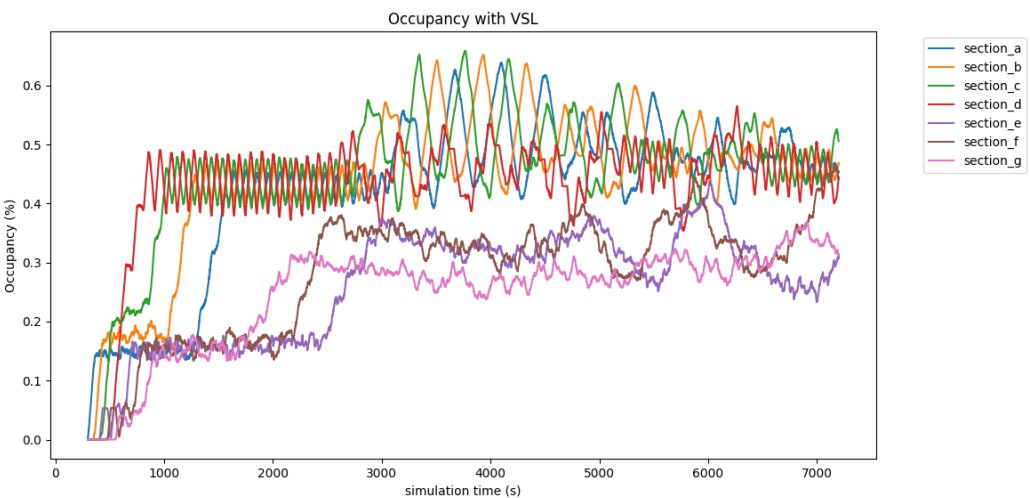

**Figure 7.** Occupancy data of different road sections during the simulation period with VSL control.

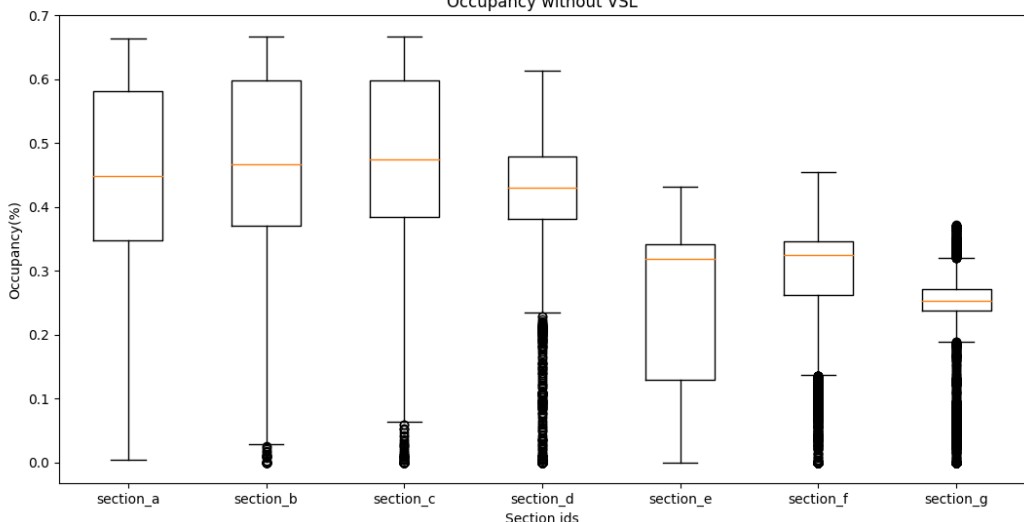

**Figure 8.** Distribution of vehicles occupancy data on each road section without VSL control.

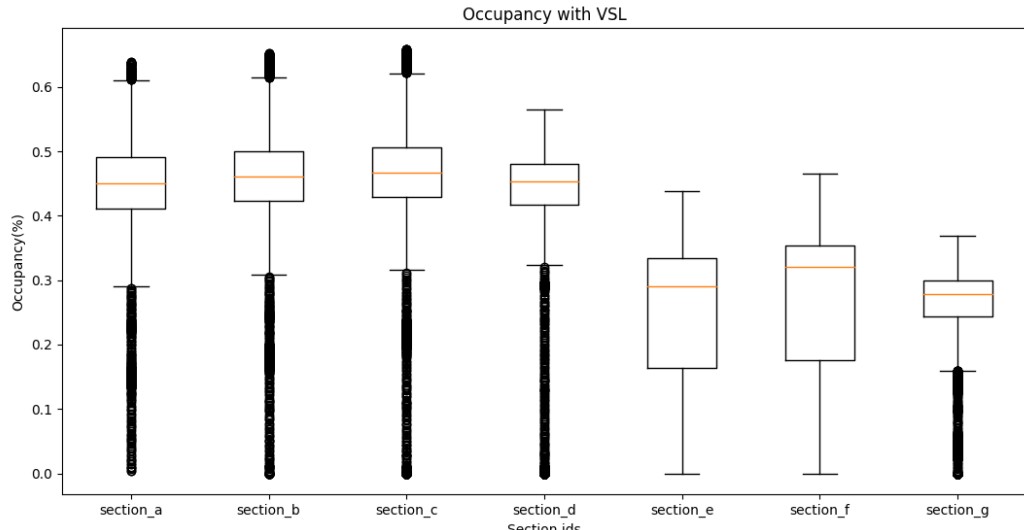

**Figure 9.** Distribution of vehicles occupancy data on each road section with VSL control.

*4.3. Sustainability Measurements*

Figure 10 shows the learning curve over 600 iterations utilizing the negative sum of the scaled carbon dioxide emissions as the reward function. As defined in Section 3, the MAPPO-based controller trends to minimize carbon dioxide emissions over the road network.

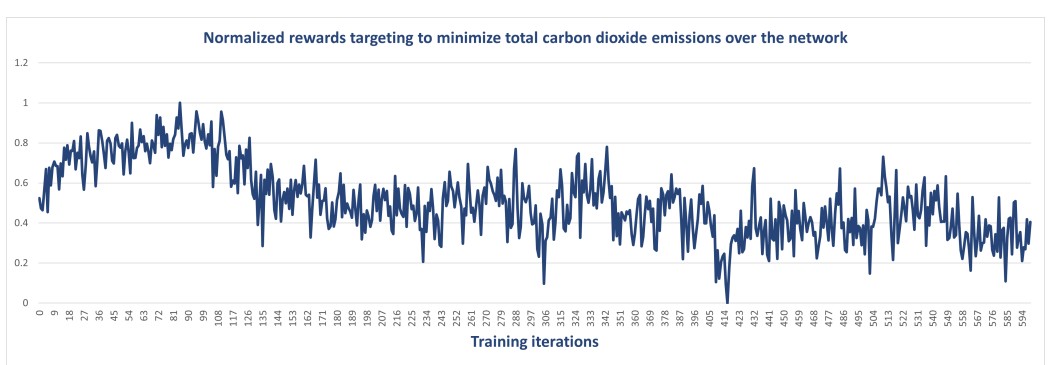

**Figure 10.** The learning curve dedicated to reducing carbon dioxide emissions over the 600 iterations.

As Figure 11 shows, the MAPPO-based VSL controller started to reduce the speed limit of all controlled sections a bit later than the controller dedicated to smoothing the traffic flow. The final speed limit chosen by the agents is 8.3 m/s for all sections. This can be explained by the innate character of the SUMO emission model. SUMO utilizes the third version of the Handbook Emission Factors for Road Transport (HBEFA 3) [53] model to calculate emissions. The homogeneous low-speed traffic flow keeps low pollutant emissions. So the agents choose the same relatively low-speed limit for all road sections.

Figure 12 shows the comparison of the total carbon dioxide emissions in kilograms without VSL control, with only single-section VSL control, and with multi-section VSL control over the simulation time with the first 300 s of warming up time. The figure verifies the performance of the multi-agent RL-based VSL controller, showing that the proposed controller performs well in terms of the sustainability metrics. Specifically, compared with the no-control case, the proposed VSL control strategy reduces the total carbon dioxide emissions of the road network by 11.2%. By only controlling the upstream section close to the bottleneck, the total carbon dioxide emissions are just reduced 0.41%. This means that compared with only adjusting the speed limit of the upstream section close to the bottleneck, our MARL-based VSL controller reduces the total carbon dioxide emissions by 10.79%.

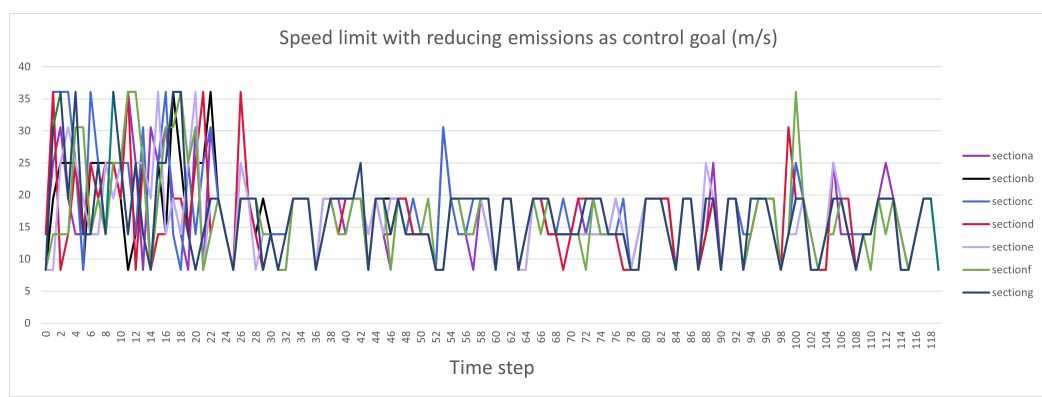

**Figure 11.** The speed limit of different road sections with reducing emissions as a control goal.

Figure 13 shows the comparison of the total waiting time in seconds without VSL control, with only single-section VSL control, and with multi-section VSL control. Compared with the single-section control case, the proposed MARL-based VSL control strategy further reduces the total waiting time through the network by 15.8%. The oscillations in the total waiting time are significantly smaller than in the uncontrolled and single-section control cases. This shows that the waiting times for vehicles tend to be similar throughout the simulation time, which demonstrate a reduction in the stop-and-go phenomenon of the vehicles.

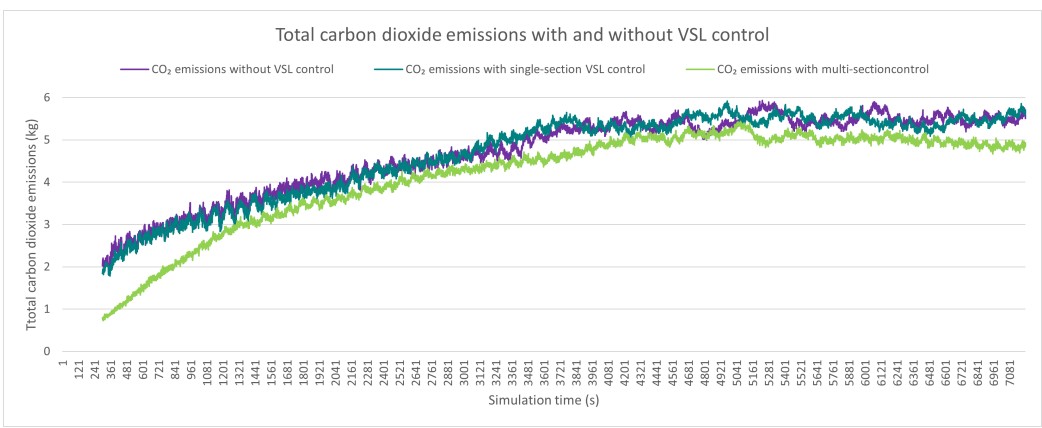

**Figure 12.** Total carbon dioxide emissions over the network with and without VSL control.

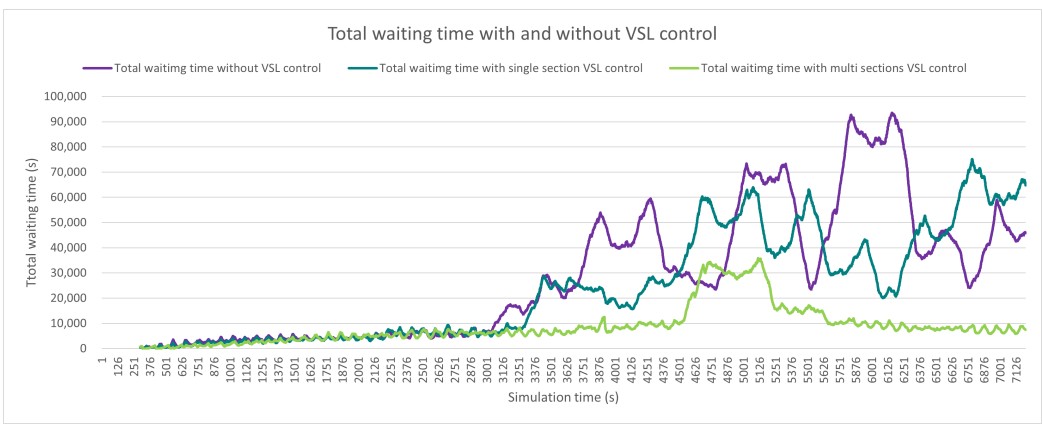

**Figure 13.** Total waiting time with and without VSL control.

Figures 12 and 13 demonstrate the superiority of our control strategy compared to the single-section VSL control. Due to the high synergism of the traffic system, single-section VSL control will lead vehicles to decelerate suddenly before the bottleneck area. Meanwhile, vehicles on the upstream section continuously drive into the bottleneck area at high speed.

High-speed cruising and low-speed congestion contribute to high road network carbon dioxide emissions. The multi-section VSL control divides the upstream of the bottleneck area into multiple deceleration sections. The control idea is to induce and control the upstream traffic flow to prevent the upstream traffic flow from accumulating quickly in the bottleneck area so that the vehicles can pass through the bottleneck area in an orderly manner. The reduction in waiting time confirms the feasibility of this control idea.

## 5. Conclusions

Different from the previous research, which only performs reinforcement learning-based variable speed limit control on the upstream section close to the bottleneck area, this paper proposes a multi-agent proximal policy optimization strategy for a multi-section variable speed limit control, which implements individual agents on all motorway sections to smooth traffic flow and reduce traffic emissions. A real-world traffic network is generated in the SUMO simulator based on the on-site historical measurement data, representing a recurrent bottleneck area when motorways transit to urban roads. The results unequivocally show improvements in the overall vehicle distribution, the total waiting time reduction, the carbon dioxide emissions reduction over the road network and how the agents apply speed limits that lead to this goal. A more homogeneous traffic flow is formed after applying the proposed MARL-based VSL control. Compared to the existing RL-based control method, which only controls the speed limit of the upstream section close to the bottleneck, our control idea outperforms. In the case of traffic performance, the total waiting time through the network is reduced by 15.8%. Regarding sustainability measurement, the total carbon dioxide emissions over the network are reduced by 10.79%.

In conclusion, the paper investigates the effect of two different controllers independently in a credible way. Limited by the iteration speed of MAPPO algorithms and the running speed of the microscopic traffic simulation, the proposed MARL-based VSL control framework is unable to realize real-time on-site traffic management. As a future work, the proposed MAPPO-based VSL control framework will be extended to multi-objective optimization to balance traffic performance and emissions by adjusting the weights of multiple reward designs on the designated road network.

**Author Contributions:** Conceptualization, X.F. and T.T.; methodology, X.F. and T.T.; case study, X.F.; validation, X.F.; writing—original draft preparation, X.F.; writing—review and editing, X.F., T.P. and T.T.; supervision, T.T.; project administration, T.T. and T.P. All authors have read and agreed to the published version of the manuscript.

**Funding:** Project no. TKP2021-NVA-02 was implemented with the support provided by the Ministry of Culture and Innovation of Hungary from the National Research, Development and Innovation Fund, financed under the TKP2021-NVA funding scheme. The research was supported by the European Union within the framework of the National Laboratory for Autonomous Systems (RRF-2.3.1-21-2022-00002).

**Institutional Review Board Statement:** Not applicable.

**Informed Consent Statement:** Not applicable.

**Data Availability Statement:** Not applicable.

**Conflicts of Interest:** The authors declare no conflict of interest.

## Abbreviations

The following abbreviations are used in this manuscript:

| | |
|---|---|
| ITS | Intelligent Transportation System |
| VSL | Variable Speed Limit |
| DRL | Deep Reinforcement Learning |
| DQN | Deep Q-Network |
| MAPPO | Multi-Agent Proximal Policy Optimization |

| | |
|---|---|
| SUMO | Simulation of Urban Mobility |
| RL | Reinforcement Learning |
| ML | Machine Learning |
| DL | Deep Learning |
| DNN | Deep Neural Networks |
| NN | Neural Networks |
| PG | Policy Gradient |
| AC | Actor–Critic |
| A3C | Advantage Actor–Critic |
| TRPO | Trust Region Policy Optimization |
| PPO | Proximal Policy Optimization |
| MARL | Multi-Agent Reinforcement Learning |
| CTDE | Centralized Training with Decentralized Execution |
| GAE | Generalized Advantage Estimation |
| TraCI | Traffic Control Interface |
| OSM | OpenStreetMap |
| TTT | Total Time Spent |
| CAV | Connected and Automated Vehicles |
| V2V | Vehicle to Vehicle |
| HBEFA 3 | Third Version of Handbook Emission Factors for Road Transport |

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
