# Peer review of "Variable Speed Limit Control for the Motorway–Urban Merging Bottlenecks Using Multi-Agent Reinforcement Learning"

_sustainability, doi:10.3390/su151411464_

Round 1

Reviewer 1 Report

The paper brings an approach that integrates control strategies to VSL considering a framework involving multi-agent methods and mobility simulation. The core of the paper seems interesting, but when I read the title and the Abstract the issues are distinct. At the title the authors translate the idea to use VSL to aid traffic congestion. However, in the Abstract the objective becomes clear, which proposes to help CO2 reduce. In my opinion the explanation used to justify the choices by authors related to the MAPPO is vague. Is necessary to bring a real contribution of the MAPPO that other types approach, not bring the same contribution. I think that equations from 1-4 require better explanation. The authors used only reference [14]to show these equations. However, the Figure 1 summary clearer all procedures used. Figure 8and 9 there a lack of the motivation on outliers. The lack of discussion section again not allow see which are the real contribution of the paper. In the conclusion section the author must show the limitations of the research.

only minor types errors

Author Response

Response to Reviewer Comments

Point: The paper brings an approach that integrates control strategies to VSL considering a framework involving multi-agent methods and mobility simulation. The core of the paper seems interesting, but when I read the title and the Abstract the issues are distinct. At the title the authors translate the idea to use VSL to aid traffic congestion. However, in the Abstract the objective becomes clear, which proposes to help CO2 reduce.

Response: Thank you for your comment. According to your remarks, we revised the Abstract to better highlight the goal of the paper. Actually, the goal is twofold. On the one hand, we want to handle the complex problem at a special but typical traffic spot, i.e., we provide a method to homogenize the traffic at the weaving area of motorway and urban roads (which obviously also means a congestion reduction). On the other hand, we also provide a method to tackle the CO2 emission problem.

Point: In my opinion the explanation used to justify the choices by authors related to the MAPPO is vague. Is necessary to bring a real contribution of the MAPPO that other types approach, not bring the same contribution.

Response: Thank you for your comment. Benchmarking the applied RL algorithm with others is the goal of our paper, i.e., we are traffic engineers hopefully providing a useful and new approach to the problem of merging traffic area control (where motorways meet slower urban roads).

To our knowledge, by now, no similar paper has been published in the traffic engineering community on this special problem.

The goal of the paper is rather to justify the applicability and performance of the control idea at this special spot of the network, and not to necessarily find the best-performing AI algorithm (as we are not AI specialists). The main purpose of applying MAPPO-based control is it is fully cooperative in contrast to human manual interventions (which is the current case, i.e., road operators manually operate the VSL signs).

Point: I think that equations from 1-4 require better explanation. The authors used only reference [14]to show these equations. However, the Figure 1 summary clearer all procedures used. Figure 8and 9 there a lack of the motivation on outliers. The lack of discussion section again not allow see which are the real contribution of the paper.

Response: Thank you for your comment. We reformulate the beginning part of section 2.2 with additional explanations of notations to let the audience better understand the algorithm.

After applying the MRAL-based VSL control, the occupancy data was more concentrated, which is exactly one of the goals of this paper. This caused more outliers compared to no control case. During the practice, the outliers usually filter out using various methods such as Tukey's fences, Z-score or standard deviation, and use percentiles to remove outliers. Instead of removing the outliers, we rather keep them to show the diversity of data.

In order to better highlight our contribution, we added two new figures to the paper, showing the comparison of the traffic performance (total waiting time through the network) and sustainability measurement (total CO2 emissions over the network), see Fig. 12 and Fig.13. Compared to existing RL-based control method which only controls the speed limit of the upstream section close to bottleneck, our control idea outperformed, in case of traffic performance, the total waiting time through the network reduced by 15.8%. Regarding sustainability measurement, the total CO2 emissions over the network were reduced by 10.79%.   

Point: In the conclusion section the author must show the limitations of the research.

Response: Thank you for your comment. We revised the conclusion part according to your comment. Limited by the iteration speed of MAPPO algorithms and the running speed of microscopic traffic simulation, the proposed MARL-based VSL control framework is unable to realize real-time on-site traffic management.

Reviewer 2 Report

The authors integrate a reinforcement learning (RL) technique, Multi-Agent Proximal Policy Optimization (MAPPO), with the Variable Speed Limit (VSL) control strategy to improve traffic efficiency and reduce vehicle occupancy on the roads and carbon dioxide emissions. The resulting control strategy is tested on the Simulator of Urban MObility (SUMO), obtaining favorable results.

The authors indicate that RL-based VSL control approaches using matured Deep Q-Network (DQN) exist. Why is there not a comparison against this technique?

Line 85. It seems like there is an extra “s”.

Line 102. “the local optimal value” might be changed to “a local optimal value”.

Lines 105 and 276. “i.e.” might be changed to “i.e.,”.

The methodology section should begin with a brief explanation of the section’s structure. Otherwise, the first lecture seems a bit chaotic. Particularly, the relevance of Section 2.2 should be emphasized.

The authors used infrastructure data from a typical recurrent bottleneck area located on the border of Budapest in Hungary. Why use this area? Is the resulting control strategy being implemented in this place?

Author Response

Thanks for your comments. Please see the attachment, which provides a point-by-point response.

Reviewer 3 Report

- Are the results averaged across multiple simulation runs with different simulation random seeds?  If yes, please state this clearly.  If no, would suggest doing this so that the results are not based on only one run of simulation - which good results may just be happened with this particular seed of simulation.

- Figure 7 (Occupancy data of different road sections during the simulation period with VSL control) and the description "The occupancy of "section_e" and "section_g" reaches the same value of 30%.": it is not conclusive that the converged value is 30% as the 30% is just one value at the end of the simulation.  To show convergence, we would need to extend the simulation longer and observe that the values stay at some particular value for some duration of simulation time.  Additionally, we can observe that section_f's value actually increases significantly towards the end of the simulation - whereas it stays relatively constant in Figure 6.  Good to explain this phenomena.

- Conclusion section: "Different from the previous research, which only performs reinforcement learning based variable speed limit control on the upstream section close to the bottleneck area ...": would be insightful doing additional experiments to compare against this approach of only controlling the upstream section close to the bottleneck area.

- By controlling the VSL on these sections, would it be just pushing the congestion to the upstream?  Good to also plot the results on "Traffic Performance Measurements" and "Sustainability Measurements" to include the upstream, i.e., the upstream section before section_a.

Author Response

Response to Reviewer Comments

Point: Are the results averaged across multiple simulation runs with different simulation random seeds?  If yes, please state this clearly.  If no, would suggest doing this so that the results are not based on only one run of simulation - which good results may just be happened with this particular seed of simulation.

Response: Thank your comment. Yes, in each iteration, a random seed is applied for the proposed control strategies to keep diversity. We state this in the manuscript in section 4.1 in line 256.

Point: Figure 7 (Occupancy data of different road sections during the simulation period with VSL control) and the description "The occupancy of "section_e" and "section_g" reaches the same value of 30%.": it is not conclusive that the converged value is 30% as the 30% is just one value at the end of the simulation.  To show convergence, we would need to extend the simulation longer and observe that the values stay at some particular value for some duration of simulation time.  Additionally, we can observe that section_f's value actually increases significantly towards the end of the simulation - whereas it stays relatively constant in Figure 6.  Good to explain this phenomena.

Response: Thank your comment. Figure 7 shows the occupancy measurements of each road section applying the optimal control, i.e., the speed limit the last iteration produces. The convergence can be observed in Figure 4. In the case of traffic performance improvement, we focus on traffic performance during a specific time period. The observation that section_f's value actually increases significantly towards the end of the simulation, is exactly what we want to achieve, i.e., by applying the proposed VSL control, a homogeneous traffic flow can be formed.

Point: Conclusion section: "Different from the previous research, which only performs reinforcement learning based variable speed limit control on the upstream section close to the bottleneck area ...": would be insightful doing additional experiments to compare against this approach of only controlling the upstream section close to the bottleneck area.

Response: Thank your comment. We added an additional experiment to compare both the traffic performance and sustainability measurement applied our VSL control framework and existing RL-based VSL control approach, which the control idea only controls the upstream road section close to the bottleneck area. The result is shown in Figure 12 and Figure 13. Compared to existing RL-based control method which only controls the speed limit of the upstream section close to bottleneck, our control idea outperformed, in case of traffic performance, the total waiting time through the network reduced by 15.8%. Regarding sustainability measurement, the total CO2 emissions over the network were reduced by 10.79%.  

Point: By controlling the VSL on these sections, would it be just pushing the congestion to the upstream?  Good to also plot the results on "Traffic Performance Measurements" and "Sustainability Measurements" to include the upstream, i.e., the upstream section before section_a.

Response: Thank your comment. The applied traffic network in the SUMO simulator is as shown by Fig. 3, i.e., section a is the starting link, so unfortunately we cannot provide any simulation results before that section. Yes, we agree with your comment that applying VSL in the study area might push congestion to upstream. However, this is an eternal question in all traffic engineering applications what is the considered managed area? There will be also a managed, protected zone which is usually assigned by the practitioners (i.e., the local traffic road operators). The same happens in the so-called “perimeter control” strategies in urban context, where of course, the protected inside urban zone is the main target to be managed in an optimal way, see for example the papers:

https://www.sciencedirect.com/science/article/pii/S0191261514001179

https://www.sciencedirect.com/science/article/pii/S1569190X22001502

Round 2

Reviewer 1 Report

there no comments

Author Response

Thank you again for reviewing our article and providing comments.

Reviewer 3 Report

- Suggest to add more discussions on why the proposed control strategy performed better than the single section control case (figures 12 and 13) - rather than just presenting the results.

Author Response

Thanks for your comment. We have revised section 4.3 accordingly.